# Patient-Centered Care for People with Depression and Anxiety: An Integrative Review Protocol

**DOI:** 10.3390/jpm11050411

**Published:** 2021-05-13

**Authors:** Lara Guedes de Pinho, Tânia Correia, Manuel José Lopes, César Fonseca, Maria do Céu Marques, Francisco Sampaio, Helena Reis do Arco

**Affiliations:** 1São João de Deus School of Nursing, University of Évora, 7000-811 Évora, Portugal; mjl@uevora.pt (M.J.L.); cfonseca@uevora.pt (C.F.); mcmarques@uevora.pt (M.d.C.M.); 2Comprehensive Health Research Centre (CHRC), 7000-811 Évora, Portugal; helenarco@ipportalegre.pt; 3Abel Salazar Institute of Biomedical Sciences, University of Porto, 4050-313 Porto, Portugal; tsp.correia@gmail.com; 4“NursID: Innovation & Development in Nursing” Research Group, CINTESIS—Center for Health Technology and Services Research, 4200-450 Porto, Portugal; fsampaio@ufp.edu.pt; 5Higher School of Health Fernando Pessoa, 4249-004 Porto, Portugal; 6Superior School of Health, Polytechnic Institute of Portalegre, 7300-555 Portalegre, Portugal

**Keywords:** anxiety, depression, patient-centered care, patient care planning, symptom assessment, patient health questionnaire

## Abstract

**Introduction**: Depression and anxiety are mental diseases found worldwide, with the tendency to worsen in the current pandemic period. These illnesses contribute the most to the world’s rate of years lived with disability. We aim to identify and synthesize indicators for the care process of the person with depression and/or anxiety disorders, based on patient-centered care, going through the stages of diagnostic assessment, care planning, and intervention. **Methods and analysis:** An integrative literature review will be conducted, and the research carried out on the following databases: MEDLINE, PsycINFO, Scopus, and Psychology and Behavioral Sciences Collection, CINAHL, Web of Science, TrialRegistry, and MedicLatina. The research strategy contains the following terms MesH or similar: “patient-centered care,” “depression,” and “anxiety.” Two independent revisers will perform the inclusion and exclusion criteria analysis, the quality analysis of the data, and its extraction for synthesis. Disagreements will be resolved by a third revisor. All studies related to diagnostic assessment, care planning, or intervention strategies will be included as long as they focus on care focused on people with depression and anxiety, regardless of the context. Given the plurality of the eligible studies, we used the narrative synthesis method for the analysis of the diagnostic assessment, the care and intervention planning, and the facilitators and barriers. **PROSPERO** registration number: CRD42021235405.

## 1. Introduction

According to data from the World Health Organization, it is estimated that, in 2015, 300 million people, that is, 4.4% of the world population, suffered from depression, and a very similar number of people would have some kind of anxiety disorder, although these health situations often occur simultaneously. The same organization identifies depression as the biggest cause of incapacity for productive activity and anxiety as the sixth biggest cause. In addition, depression is the mental disorder with the greatest responsibility in the number of suicide deaths, occurring about 800 thousand per year [1].

These numbers tend to worsen due to the current COVID-19 pandemic since the evidence shows that it is having a strong negative impact on people’s mental health, regardless of gender, group, or region, manifesting in depression and anxiety [2,3,4,5,6,7,8]. Recent data reveal that the prevalence of depression in the regions most affected by the pandemic is three times higher than it was in the general population in 2015 (15.97% vs. 4.4%). In turn, anxiety is four times higher (15.15% vs. 3.6%) [1,9].

Although depressive and anxiety disorders were already very prevalent before the pandemic crisis, only a small percentage, about 50%, of patients received adequate treatment. Among the identified barriers, gaps in communication and coordination among care organizations stand out, resulting in their lack of integration as a whole, an example of which is post-hospital discharge monitoring [10].

It is therefore too evident the need to develop surveillance, prevention, and intervention programs during and after this global crisis [9]. For this, a timely and adequate diagnostic evaluation is necessary, followed by care planning. Regarding care planning strategies, in 2001, the Institute of Medicine identified the importance of the centrality of care in the person as one of the six elements for high-quality care [11]. This recommendation implies the provision of care consistent with the values, needs, and wishes of the patients and an example of this is their involvement in discussions and decisions in their health process [12]. This approach intends that the provision of care has the objective of establishing a therapeutic relationship with the person focused on their needs, thus promoting their training [13]. The individual care plan (PIC) is a tool that focuses on patient-centered care, allowing an individual and holistic approach [14]. This, in turn, can encompass, for example, case management which is a patient-centered approach aimed at systematic monitoring and support of patients by a case manager [10].

A systematic review of the literature that analyzed the treatment preferences of patients with depression and anxiety disorders concluded that patients consider process and cost attributes more important than outcome attributes [15]. However, we found no literature review that addresses the patient care process (diagnostic assessment, care planning, and intervention) of people with depression and/or anxiety disorder, based on patient-centered care. Thus, the current proposed review will fill this gap in knowledge.

Patient-centered care planning strategies have important potential in the treatment of disorders such as anxiety and depression. This, in turn, should encompass pharmacological treatment and a set of activities that promote mental health through the management of self-care by the individual.

Thus, we intend to conduct an integrative literature review that aims to map studies or theoretical approaches about the care process of the person with depression and/or anxiety disorders, synthesizing indicators from the various stages of the patient care process, i.e., assessment, planning, and intervention.

### 1.1. Objective

We aim to identify and synthesize indicators for the care process of the person with depression and/or anxiety disorders, based on patient-centered care, going through the stages of diagnostic assessment, care planning, and intervention.

### 1.2. Review Questions

This review will be conducted in the sense to answer the following questions:

What are the patient-centered care strategies used in the assessment of the person with depression and/or anxiety disorder?

What are the patient-centered care strategies used in planning care for the person with depression?

What are the patient-centered care strategies used in the intervention for the person with depression and/or anxiety disorder?

## 2. Methods and Analysis

This protocol was developed according to the Preferred Reporting Items for Systematic Reviews and Meta-Analyses (PRISMA) Protocols Statement [16]. It was registered with the International Prospective Register of Systematic Reviews (PROSPERO; registration number CRD42021235405).

Considering the scope of this study is very specific and not yet very explored, the current evidence will be verified as characterized by a vast methodological diversity. Therefore, we choose to incorporate in this review, studies conducted with resources using quantitative, qualitative, mixed methodologies and theoretical approaches.

This protocol was developed in January 2021, and it is intended that the respective review is completed by the end of June 2021.

### 2.1. Eligibility Criteria

The way to guarantee the rigor and systematization specific to this type of study, eligibility criteria were defined as follows.

#### 2.1.1. Population

In relation to study participants, inclusion criteria are patients with a medical diagnosis of depression and/or anxiety disorder, regardless of the state of evolution. For example, we will include studies in which the participants have acute or chronic depression, or depression coexisting with another pathology because we want to map as much information as possible about patient-centered care pathways since they are still scarce in the scientific literature. Studies in which participants have not been previously diagnosed with depression or anxiety disorders will be excluded.

#### 2.1.2. Intervention

The current literary review will include studies about:The assessment of people diagnosed with depression and/or anxiety disorders, be it from assessment instruments or other types of evaluation to assess the results of care;Patient-centered care planning strategies for people diagnosed with depression and/or anxiety disorders, with the objective to promote mental health, to improve the quality of care, to improve patient satisfaction, to prevent and/or reduce complications of the mental health state related to the referred diagnostics;Intervention strategies that aim to implement patient-centered care.

Examples of possible strategies to include are vigilance, promotion of health, health education, psychoeducation, coaching amongst other programs of mental health that may appear in health care, school, homes, and other contexts.

#### 2.1.3. Comparison

Studies with or without comparative groups will be included in this review.

#### 2.1.4. Primary Outcome

The primary outcome considered in this review will be the change or non-worsening of the state of mental health, functioning, and well-being that can be assessed in general or specific for the diagnoses of depression and anxiety. The data can be of a quantitative nature, such as averages, measures of prevalence or incidence, frequencies, or of a qualitative nature, such as self-reported wellness and satisfaction of care. Theoretical studies may also be included according to the methodology of integrative reviews [17]. 

#### 2.1.5. Secondary Outcomes

The secondary outcome considered in this review will be patient satisfaction and the quality of care, identified in the implementation of patient-centered care.

#### 2.1.6. Study Design

This integrative review shall include empiric quantitative observational or experimental primary studies, qualitative studies, mixed studies, and theoretical studies.

#### 2.1.7. Context

All studies related to the diagnostic assessment, the planning of care, or intervention strategies focused on individuals with depression and anxiety, regardless of the age group in any geographic area, regardless of context (community, culture, or specific environment), will be included in this review.

### 2.2. Search Strategy

#### 2.2.1. Data Sources

In the research strategy, it is intended to carry out a comprehensive bibliographic search and the databases to be consulted will be: MEDLINE, PsycINFO, Scopus, and Psychology and Behavioral Sciences Collection, CINAHL, Web of Science, TrialRegistry, and MedicLatina.

#### 2.2.2. Search Terms

The research will include the combination of four key concepts according to Medical Subject Headings (MeSH) terms patient-centered care, depression, and anxiety, in the title and abstract. In this case, the search phrase could be the following:

((“Patient Care Plan*”) OR (“Patient-Centered Care”)) AND ((Depression) OR (“Depressive Disorder”) OR (Anxiety))

Other keywords can also be used if necessary, such as patient satisfaction or health education.

The research strategy will be adapted in accordance with each data bank and will be restricted to the last 10 years, i.e., from 2011 to 2021 in the English, Portuguese, French, German and Italian languages.

### 2.3. Data Collection and Analysis

#### 2.3.1. Selection of Studies

The following study of research in each database will be exported into Mendeley and the duplicates will be removed.

To minimize bias, two reviewers will independently assess the inclusion of the studies by reading the title, abstracts, and keywords and excluding those that do not fit the inclusion criteria in this review. The third reviewer should be consulted in case of disagreements or doubts. Afterward, we proceed to the assessment of the complete texts. To present this selection process, the PRISMA flowchart will be presented with the triage results in its different stages.

#### 2.3.2. Data Extraction

In the data extraction phase, initially, a descriptive evaluation of each study will be carried out using the extraction instrument designed to extract information according to the research questions. The information to be presented will be the objective of the study, participants, diagnostic assessment instruments, care planning strategies, intervention, results, and main conclusions.

The extraction of data will be conducted by the same two reviewers independently and any doubt or disagreement will be resolved by consulting the third reviewer.

#### 2.3.3. Quality Appraisal

Since this is a review that intends to integrate quantitative, qualitative, and mixed methods studies, the “mixed methods assessment tool” will be used to assess the quality of the selected studies [18]. Once again, this step will be carried out by the same two reviewers independently and any disagreement with the evaluation of the quality of the studies must be resolved once again using the third reviewer.

The result of the evaluation of the quality of each study will be presented and these data do not represent an inclusion/exclusion criterion, so all studies selected up to this stage will be included [17]. In this way, it will be possible to perceive the quality of the evidence produced within the scope of this review. Studies of low quality will contribute less to the analytic process.

#### 2.3.4. Strategy for Data Synthesis

As an integrative review that will include studies with several methodologies, the synthesis and analysis of the results will be of a narrative nature, structured as to answer the questions presented regarding the investigation. Diagnostic assessment strategies, planning of care focused on individuals with depression and/or anxiety disorders and implementation of the intervention will be presented according to the strategic nature, objectives, and study and target population scenarios (Whittemore and Knafl, 2005).

Patient and public involvement: There is no involvement of patients or the public in the design and development of the respective review.

Ethics and dissemination: Since only the data considered secondary will be analyzed, the ethical approval of this study is not necessary. This scientific paper is an integrative review protocol, in which the data has not yet been extracted or analyzed. The results will be disseminated through publications subject to peer review.

## 3. Discussion

Current evidence on the incidence of anxiety and depression has an increasing tendency associated with high levels of disability and other complications at various levels, such as social, economic, or family. Therefore, the early and effective diagnostic assessment for the development of planning of care and intervention is crucial as a response to this global phenomenon.

It is consensual, among the various organizations and bodies responsible for issuing guidelines for health care, that planning of care should be centered on the person as a guideline. If this recommendation is important in general care, in the context of mental health it is especially relevant.

In this sense, with this protocol, we intend to gather the conditions to start the review research in order to know the current evidence on diagnostic assessment, planning of patient-centered care, and intervention strategies for people with depression and anxiety.

With the elaboration of this protocol, we intend to guarantee the rigor, clarity, and quality of the process in order for it to be systematic. An example of it is the involvement of two reviewers in the multiple stages of identification and selection of studies and also the research in different databases.

As a result of this integrative review, we believe that we can contribute to the knowledge on the subject, identify guidelines for evidence-based practice, as well as perceive potential guidelines for future investigations.

## 4. Strengths and Limitations of This Study

This planned integrative review will use a detailed search strategy focusing on identifying ways of diagnostic assessment, care planning, and implementation of intervention strategies for people with depression and anxiety, based on patient-centered care models;The research strategy will be adapted according to each data bank and will be restricted to the last 10 years, i.e., from 2011 to 2021 in the English, Portuguese, French, German and Italian languages;This review will include a diversity of study designs including studies using quantitative, qualitative, and mixed methodologies;Since this is a review that aims to integrate quantitative, qualitative, and mixed methods studies, the “mixed methods assessment tool” will be used to assess the quality of the selected studies.

## Data Availability

Not applicable.

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
