# Peer review of "Patient-Centered Care for People with Depression and Anxiety: An Integrative Review Protocol"

_jpm, 2021, doi:10.3390/jpm11050411_

Round 1

Reviewer 1 Report

Thank you for the opportunity to review this Review Protocol, describing a  proposed systematic integrative review on patient-centered care models for people with Depression and Anxiety.

GENERAL COMMENTS

 - I have three concerns with the proposed review which I feel should be addressed.

  1. The authors have not sufficiently justified the need for the review in terms of: 1) what Reviews related to this topic already exist in the literature; 2) what the gaps are in these previous reviews; and 3) how the current proposed review will fill this gap in knowledge.
  2. The scope of review concerns me, as it does not appear to be limited to any particular setting. I wonder how the authors will deal with different diagnostic approaches in different settings - e.g. patients presenting to a private psychologist for a primary mental health concern (diagnosed by a psychologist and where the BDI tool might be used) vs patients in hospital for primarily medical condition such as cancer who are subsequently screened for anxiety and depression (diagnosed by a nurse where the HADS might be used) vs people in the general population who are screened (potentially not diagnosed by a health care professional at all and where the K10 might be used). I feel more information is needed regarding how screening in different settings, and screening by different health care professionals will be synthesised in the review.
  3. Finally, treatment pathways can vary substantially depending on whether depression and anxiety are acute or chronic, whether it is the first diagnoses, whether the depression coexists with another condition such as PTSD, etc. More information on exclusion criteria is needed to make the scope of the review clear.

Author Response

Dear reviewer,

Firstly, we would like to thank you very much for your analysis and for the opportunity to improve our paper. We found your recommendations extremely useful and we are sure they helped improve the overall quality of the manuscript.

We tried to address all your recommendations and to give response to all your comments:

  1. Thank you very much for your comment. We add this information in the introduction.
  2. We will only include studies in which the participants have a medical diagnosis of depression and/or anxiety disorder. All situations that you mention will be excluded. We add this information to the inclusion and exclusion criteria of the protocol.

  3. We add more information to the inclusion and exclusion criteria of the protocol.

Reviewer 2 Report

Thank you for the opportunity to review this interesting manuscript.  Overall the details were presented coherently to approach the aim of the proposed study. I have a few minor comments for the authors to consider:

  • Introduction:
    • Perhaps the authors could clarify if they were interested in the ‘effectiveness’ of the strategies, instead of just the availability of strategies? It would be good to clarify so in Section 1.2. Effectiveness links well to the primary outcome stated in the methods section.
  • Methods:
    • Line 136: Please include a brief example of qualitative measure (e.g. self-reported well, no sadness of photographs, etc?)
    • Please consider including TrialRegistry as another data source. It could be region-based if preferred. Nevertheless, this update needs to be reflected in the revised Abstract, and section 2.2.1.
    • Some examples noted from line 127 to 129 (Section 2.1.2) could be added as keywords for the search (Section 2.2.2). This would help to manage the scope of the search.
    • The primary outcome was ‘change’ (over certain period). It would be good to exclude studies who had no follow-up period (or certain time that the research team deemed too short or too long), and note this in the protocol, either as a standalone paragraph of exclusion criteria (may also exclude studies who did not use standardised diagnoses of depression or anxiety), or note the exclusion criteria as another few sentences in Section 2.3.1, line 169).
    • Section 2.3.3, line 185-186: Please clarify if the 2 independent reviewers would be the same 2 independent reviewers who selected and reviewed the studies as noted in Section 2.3.1 and Section 2.3.2.
    • Section 2.3.3: Please indicate what would happen to the studies assessed as having poor quality.
    • Section 2.3.4: Whilst the authors anticipated that it will be very difficult (line 200) to conduct a meta-analysis, this may not be the case. It is important to outline the steps for meta-analyses in a protocol, in order to lead the analyses should there be no huge heterogeneity amongst the shortlisted studies. Meta-analysis is important as the authors aimed to “contribute to the knowledge on the subject, identify guidelines for an evidence-based practice, as well as perceive potential guidelines…” (line 227-228).
    • The authors may also wish to register the revised protocol on a repository, such as Prospero, so that this work would not be replicated by other research team(s). 

Author Response

Dear reviewer,

Firstly, we would like to thank you very much for your analysis and for the opportunity to improve our paper. We found your recommendations extremely useful and we are sure they helped improve the overall quality of the manuscript.

We tried to address all your recommendations and to give response to all your comments.

Introduction:

  • Perhaps the authors could clarify if they were interested in the ‘effectiveness’ of the strategies, instead of just the availability of strategies? It would be good to clarify so in Section 1.2. Effectiveness links well to the primary outcome stated in the methods section.

We have made changes in the introduction and in point 1.2 to clarify this. We do not intend to evaluate the effectiveness of interventions, but rather to map what exists about the process of caring for the person with depression and/or anxiety based on patient-centered care, whether in assessment, care planning, or intervention. In fact, that was not very explicit. Hopefully, it is now clear to the readers. We think we have clarified that now.

Methods:

  • Line 136: Please include a brief example of qualitative measure (e.g. self-reported well, no sadness of photographs, etc?)

We add this information.

  • Please consider including TrialRegistry as another data source. It could be region-based if preferred. Nevertheless, this update needs to be reflected in the revised Abstract, and section 2.2.1.

Thank you for your suggestion. We add this information.

  • Some examples noted from line 127 to 129 (Section 2.1.2) could be added as keywords for the search (Section 2.2.2). This would help to manage the scope of the search.

We add this information.

  • The primary outcome was ‘change’ (over certain period). It would be good to exclude studies who had no follow-up period (or certain time that the research team deemed too short or too long), and note this in the protocol, either as a standalone paragraph of exclusion criteria (may also exclude studies who did not use standardised diagnoses of depression or anxiety), or note the exclusion criteria as another few sentences in Section 2.3.1, line 169).

After further review of the protocol and discussion among the authors, we realized that the outcomes we intended to map were not quite explicit and made slight changes. Please consider these new changes.

  • Section 2.3.3, line 185-186: Please clarify if the 2 independent reviewers would be the same 2 independent reviewers who selected and reviewed the studies as noted in Section 2.3.1 and Section 2.3.2.

Yes, the two independent reviewers are the same. We add this information in 2.3.2 and 2.3.3.

  • Section 2.3.3: Please indicate what would happen to the studies assessed as having poor quality.

We add this information in 2.3.3.

  • Section 2.3.4: Whilst the authors anticipated that it will be very difficult (line 200) to conduct a meta-analysis, this may not be the case. It is important to outline the steps for meta-analyses in a protocol, in order to lead the analyses should there be no huge heterogeneity amongst the shortlisted studies. Meta-analysis is important as the authors aimed to “contribute to the knowledge on the subject, identify guidelines for an evidence-based practice, as well as perceive potential guidelines…” (line 227-228).

Thank you for this pertinent comment. We have checked the whole protocol and we understand that maybe it was not clear that it is an integrative literature review and not a systematic review, but that was our oversight. So first we will do an integrative review that includes all kinds of studies and theorical studies. If we find many experimental studies, we will choose to do a systematic review of the literature afterwards with only interventions. In this first phase we just want to map everything that exists. We will update the protocol accordingly. We hope you will understand.

  • The authors may also wish to register the revised protocol on a repository, such as Prospero, so that this work would not be replicated by other research team(s). 

We have already registered the protocol at Prospero and we add this information at the beginning of the Methods and analysis section.

Thank you very much for your pertinent revision.